Citation: *Molecular Systems Biology* 9:702
www.molecularsystemsbiology.com

# Design of orthogonal genetic switches based on a crosstalk map of σs, anti-σs, and promoters

Virgil A Rhodius[1,6], Thomas H Segall-Shapiro[2,6], Brian D Sharon[3], Amar Ghodasara[2], Ekaterina Orlova[1], Hannah Tabakh[1], David H Burkhardt[3], Kevin Clancy[4], Todd C Peterson[4], Carol A Gross[1,5,*] and Christopher A Voigt[2,*]

[1] Department of Microbiology and Immunology, University of California San Francisco, San Francisco, CA, USA, [2] Department of Biological Engineering, Synthetic Biology Center, Massachusetts Institute of Technology, Cambridge, MA, USA, [3] Graduate Group in Biophysics, University of California San Francisco, San Francisco, CA, USA, [4] Synthetic Biology Research and Development, Life Technologies, Carlsbad, CA, USA and [5] Department of Cell and Tissue Biology, University of California San Francisco, San Francisco, CA, USA
[6] These authors contributed equally to this work
* Corresponding author. CA Gross, Department of Microbiology and Immunology, University of California San Francisco, 600 16th Street, San Francisco, CA 94158, USA. Tel.: + 1 415 476 4161; Fax: + 1 415 514 4080; E-mail: cgrossucsf@gmail.com or CA Voigt, Department of Biological Engineering, Synthetic Biology Center, Massachusetts Institute of Technology, 500 Technology Square NE47-277, Cambridge, MA 02139, USA. Tel.: + 1 617 324 4851; E-mail: cavoigt@gmail.com

Cells react to their environment through gene regulatory networks. Network integrity requires minimization of undesired crosstalk between their biomolecules. Similar constraints also limit the use of regulators when building synthetic circuits for engineering applications. Here, we mapped the promoter specificities of extracytoplasmic function (ECF) σs as well as the specificity of their interaction with anti-σs. DNA synthesis was used to build 86 ECF σs (two from every subgroup), their promoters, and 62 anti-σs identified from the genomes of diverse bacteria. A subset of 20 σs and promoters were found to be highly orthogonal to each other. This set can be increased by combining the − 35 and − 10 binding domains from different subgroups to build chimeras that target sequences unrepresented in any subgroup. The orthogonal σs, anti-σs, and promoters were used to build synthetic genetic switches in *Escherichia coli*. This represents a genome-scale resource of the properties of ECF σs and a resource for synthetic biology, where this set of well-characterized regulatory parts will enable the construction of sophisticated gene expression programs.
*Molecular Systems Biology* **9**: 702; published online 29 October 2013; doi:10.1038/msb.2013.58
*Subject Categories:* metabolic and regulatory networks; synthetic biology
*Keywords:* compiler; genetic circuit; part mining; synthetic biology; systems biology

## Introduction

Bacterial sigma factors (σs), the promoter recognition subunits of RNA polymerase (RNAP), are modular proteins with domains that recognize DNA sequences in the − 10 and − 35 regions of their target promoters (Hook-Barnard and Hinton, 2007). In addition to the housekeeping σs (e.g., $\sigma^{70}$ in *E. coli*) that recognize the thousands of canonical promoters essential for growth, bacteria have a variable number of stress-activated alternative σs that direct RNAP to distinct promoter sequences. This enables cells to express multiple genes associated with a particular developmental state or stress response (Gruber and Gross, 2003) and to execute complex gene expression dynamics that implement temporal control and serve as developmental checkpoints (Chater, 2001). For example, spore formation in *B. subtilis* requires a cascade of five σs ($\sigma^H \rightarrow \sigma^F \rightarrow \sigma^E \rightarrow \sigma^G \rightarrow \sigma^K$) (Stragier and Losick, 1990). σs can be embedded in complex webs of partner swapping networks, including anti-σs, which physically block σs from interacting with RNAP (Helmann, 1999; Campbell *et al*, 2008; Staroń *et al*, 2009), and anti-anti-σs. Such feedback loops and protein–protein interactions generate more complex dynamics

for integrating many environmental and cellular signals (Marles-Wright and Lewis, 2007).

Extracytoplasmic function (ECF) σs are the smallest and simplest alternative σs, as well as the most abundant and phylogenetically diverse (Helmann, 2002; Staroń *et al*, 2009). Possessing just the two domains that bind the promoter − 10 and − 35 regions (Gruber and Gross, 2003) (Figure 1A), they provide cells with a highly modular means to react to their environment (Lonetto *et al*, 1994; Staroń *et al*, 2009), often responding to a signal through the action of an anti-σ. ECF σs usually autoregulate their own expression and that of their anti-σ (Rouvière *et al*, 1995; Rhodius *et al*, 2005). This organization can lead to diverse dynamical phenomena, including ultrasensitive bistable switches and pulse generators (Voigt *et al*, 2005; Locke *et al*, 2011; Tiwari *et al*, 2011). Moreover, promoters of an ECF σ are highly conserved, facilitating identification, modeling, and rational design (Staroń *et al*, 2009; Rhodius and Mutalik, 2010). Promoter specificity also results in a large dynamic range of output, where the OFF state is very low in the absence of the σ and the ON state produces a high level of expression.

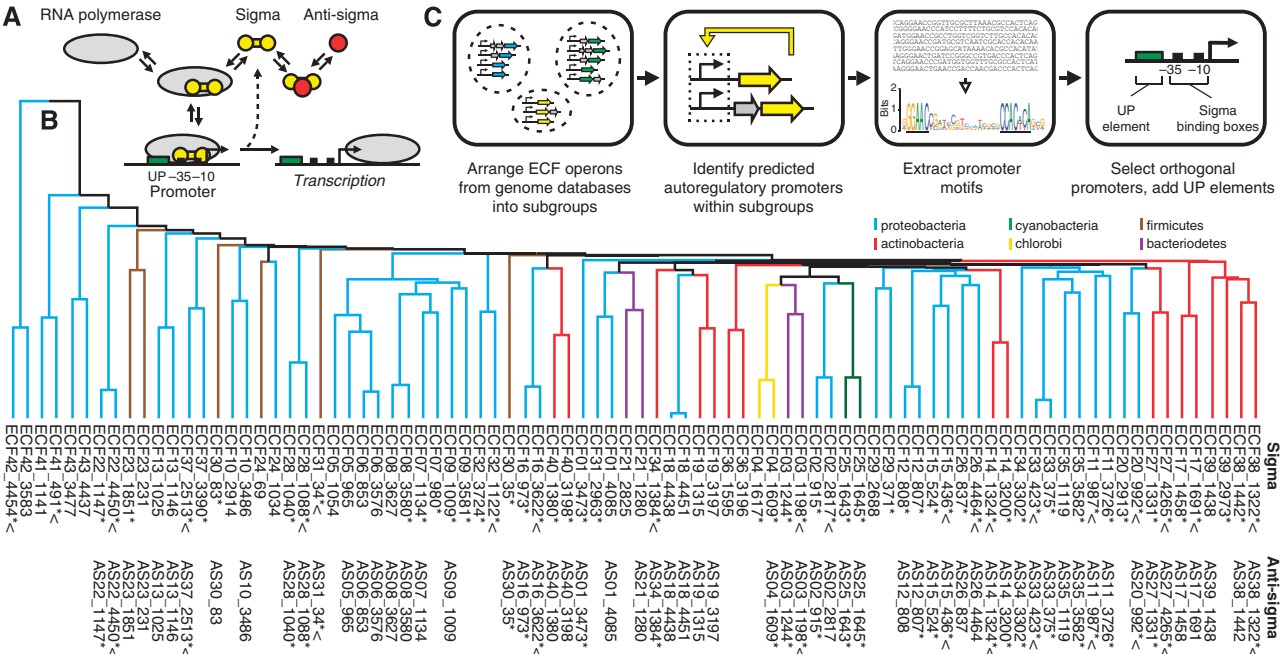

**Figure 1** The strategy for the genomic mining of ECF σs, anti-σs, and promoters is shown. (**A**) σs recruit core RNAP to promoters; a function that is inhibited by the anti-σ. σs have a two-domain structure that binds to the − 10 and − 35 regions of the target promoter. (**B**) The complete libraries of 86 synthesized σs (top row) and their 62 cognate anti-σs (bottom row) are shown organized as a phylogenetic tree. Asterisks indicate active σs (>5-fold activation) or anti-σs (>2-fold repression). Carets indicate σs or anti-σs that appear in the final orthogonal sets. All σs in the library are named ECF**XX**_**YYYY**, where 'XX' denotes the ECF subgroup, and 'YYYY' denotes the unique σ ID given by Staroń *et al* (2009). The anti-σs were named AS**XX**_**YYYY**, where 'XX' and 'YYYY' denote the ECF subgroup and unique ID of the cognate σ. Consequently, cognate σ/anti-σ pairs have the same numbering (e.g., ECF11_987 and AS11_987). (**C**) For each σ, target promoters are identified through a process of computational search, selection, and design. The first step involves the organization of the ECF operons according to the subgroups defined by Mascher and co-workers (Staroń *et al*, 2009).

Their aggregate properties suggest that ECF σs are an underused, but potentially valuable resource for implementing synthetic programs of gene expression for applications in biotechnology. Individual genetic circuits have been constructed using ECF σs to implement memory and timer functions (Chen and Arkin, 2012; Shin and Noireaux, 2012). Such circuits can be connected to implement programmable control over metabolic pathways and cellular functions (Solomon *et al*, 2012; Temme *et al*, 2012; Zhang *et al*, 2012). The size and sophistication of such circuits has been growing, but have been limited by a lack of regulatory parts that are orthogonal; that is, can be simultaneously used in a circuit without interference (Clancy and Voigt, 2010). In the case of ECF σs, non-orthogonality could arise if two σs activate each other's cognate promoter, bind to the same anti-σ, or influence each other via their sharing of RNAP holoenzyme (Grigorova *et al*, 2006; Del Vecchio *et al*, 2008).

There is evidence that there may be a large reservoir of potentially orthogonal ECF σs present in the sequence databases. Currently, there are 19 314 unique ECF σs annotated in the MiST database (Ulrich and Zhulin, 2007). Bioinformatic analysis of the sequence relationships among ∼2700 ECF σs by Mascher and colleagues identified 43 phylogenetically distinct ECF σ subgroups. These are thought to have similar promoter binding sequences within subgroups, but with a significant variation between subgroups (Staroń *et al*, 2009). This sequence diversity implies that the ECF σ family could be an ideal source to identify orthogonal regulators that could be used together to build circuits within a single cell.

In this work, we mined this diversity using DNA synthesis and rigorously mapped the crosstalk between σs and promoters to identify a core orthogonal set. First, two representatives were identified from each subgroup to build a library of 86 σs. Then, a promoter model for each subgroup was parameterized and used to scan bacterial genomes to identify putative promoters. Using this approach, 26 functional promoters were identified and the $26 \times 86 = 2236$ possible crossreactions screened. A subset of 20 σ:promoter pairs were found that exhibited minimal crosstalk. A similar approach was applied to identify anti-σs that downregulated only their cognate σ. These σs, promoters, and anti-σs were used to build a set of threshold-gated switches. Each switch contained a promoter input and output, and their threshold and cooperativity was tuned by modulating the expression level of the anti-σ. These findings demonstrate the powerful approach of genome mining for developing new resources for synthetic biology, and highlight the transferability of these regulators from diverse genomes (i.e., they still retain function in a new host with minimal re-engineering).

## Results

### Model-guided mining of ECF σs and their cognate promoters from genome sequences

Part mining refers to the application of bioinformatics and DNA synthesis to physically access large sets of parts from sequence databases. This approach has been applied to build

libraries of enzymes (Bayer *et al*, 2009), transporters (Dunlop *et al*, 2011), and simple regulatory parts (e.g., terminators) (Cambray *et al*, 2013; Chen *et al*, 2013). Here, we applied this approach to build a phylogenetically diverse library of 86 σs, which comprises 2 σs from each ECF subgroup (Figure 1B; Supplementary Information II.A; Supplementary Table S1.1). Transcriptional regulatory proteins present a particular challenge for mining, as it is necessary to determine their target DNA sequences to assess their functionality. In our case, this information is required to design a responsive promoter. To this end, we developed a computational approach that identified native promoters for each subgroup and used their sequences to build a subgroup-specific promoter model. These models were then used to screen the identified native promoters to predict whether they would be specifically activated by σs from their own subgroup without crossreacting with σs from other subgroups (i.e., orthogonal).

Native promoters were identified by exploiting the fact that most of the ECF σs autoregulate, in other words, they are transcribed from an upstream promoter recognized by the ECF σ itself (Helmann, 2002; Staroń *et al*, 2009). Consequently, promoter motifs were found by searching for overrepresented conserved motifs in the regulatory regions upstream of the σ genes in each subgroup. Using this information, Mascher and co-workers identified motifs for 18/43 σ subgroups (Staroń *et al*, 2009). Using an automated procedure, we extracted all regulatory regions upstream of the σs and their putative operons from a set of 329 genomes (Supplementary Information I.A). Conserved promoter-like motifs were identified from the upstream sequences using BioProspector, which can search for two sequence blocks (i.e., the −10 and −35 regions) connected by a variable spacer (Liu *et al*, 2001). This approach confirmed and expanded the motifs identified by Mascher and co-workers (Staroń *et al*, 2009). Our combined efforts identified 706 promoters and 29 unique promoter motifs in the 43 ECF σ subgroups. We constructed promoter models for the promoter motifs based on position weight matrixes (PWMs) (Staden, 1984) for each ECF subgroup (Rhodius and Mutalik, 2010) and a spacer penalty for suboptimal motif spacing (Figure 2 and Supplementary Figure S1). The six Fec-I like subgroups (ECF05 − 10) were excluded from this analysis as they do not autoregulate (Braun *et al*, 2003; Staroń *et al*, 2009).

Using these promoter models, we scored all 706 promoter sequences for orthogonality and found that most promoters are highly orthogonal, with remarkably little crosstalk across subgroups (Supplementary Figure S2a). Surprisingly, the −10 and −35 sequences alone show considerably less

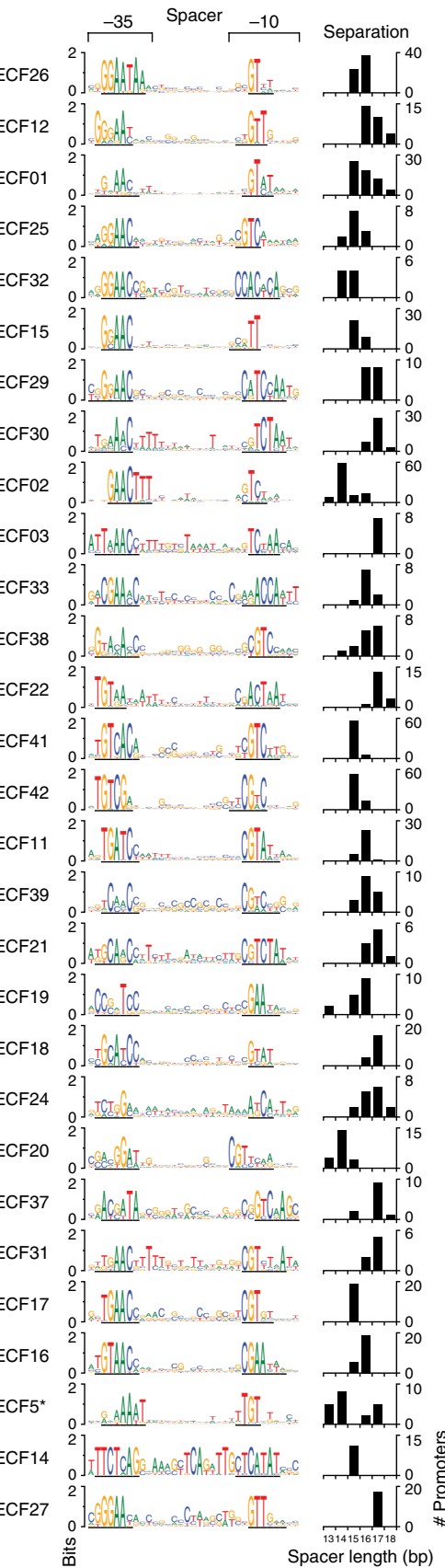

**Figure 2** Promoter models are shown for 29 ECF σ subgroups. The models contain a sequence logo illustrating the −35/−10 motifs and intervening spacer sequence. The exact −35 and −10 sequences identified by BioProspector (Liu *et al*, 2001) are underlined underneath each sequence logo. The bar chart histograms illustrate the number of promoters with different length distances between underlined the −35 and −10 motifs. The promoters were organized vertically to cluster similar −35 and −10 motifs, as determined by eye. The bottom three promoter models (ECF5*, ECF14, and ECF27) represent promoters that were not found to be active ( > 5-fold activation) in our tests. Promoter model ECF5* represents the model for subgroups 5–10.

orthogonality than the entire promoter (Supplementary Figure S2b and c), indicating that high specificity is achieved by combining both promoter regions. It also implies that new promoter specificities can be achieved by swapping the protein domains that bind the − 10 and − 35 promoter regions.

We designed candidate orthogonal promoters for each of the 29 ECF σ subgroups by extracting native promoter sequences from − 60 to + 20 based on the predictions of the promoter models across all 706 promoters. The promoters selected were predicted to score highly against their cognate σ and poorly against other σs. Preference was given to promoters that were identified immediately upstream of their cognate σ. Additionally, to prevent crosstalk with the other *E. coli* σs, we screened out promoters with sequences similar to those recognized by *E. coli* $\sigma^{70}$ and FecI (Supplementary Information I.B, I.D, and II.A). This was a particularly important step for promoters from AT-rich genomes that often contain $\sigma^{70}$-like promoter sequences. Promoters from GC-rich genomes were often found to be non-functional in our *E. coli* host. This was corrected by replacing the − 35 to − 60 region with a synthetic A/T-rich UP element designed to enhance promoter recognition by the α subunits of RNAP (Gourse *et al*, 2000; Rhodius *et al*, 2012) (Figure 1C; Supplementary Figure S3).

## Measurement of activity and crosstalk between the σ and promoter pairs

A multi-plasmid system was designed to measure promoter activity and orthogonality (Figure 3A; Supplementary Figure S21). This approach has the advantage of providing tight control of σ expression from a T7 RNAP promoter, which is useful both to reduce background when uninduced, and to reduce potential problems when synthesizing the σ genes. It also allows for the rapid transformation of different combinations of σs, anti-σs, and promoters to measure activity and orthogonality.

The 86 σ genes, optimized for expression in *E. coli* (Raab *et al*, 2010), were obtained via DNA synthesis and placed under

the control of a promoter responsive to T7 RNAP on a pBR322 plasmid. The expression of an attenuated T7 RNAP (Temme *et al*, 2012) was controlled from a separate pIncW plasmid under an isopropyl β-D-1-thiogalactopyranoside (IPTG) inducible control. Promoter mining yielded 29 promoters designed to be specific to different subgroups. These promoters were placed on a pSC101 plasmid driving sfGFP expression. Each promoter was screened against its cognate σ to measure the steady-state response function. After some promoters were modified to include a synthetic UP region (Supplementary Information I.F), we found that 18 promoters were functional with σs from their cognate ECF subgroups (Supplementary Figure S3). The dynamic range of the responses ranged from 12-fold (P25_up4311) to 480-fold (P16_3622). The level of leakiness of each promoter in the absence its σ also varied approximately 10-fold. Despite not being active in the initial screen, the promoters from the remaining 11 subgroups were retained in case they were activated by σs from different subgroups.

The toxicity of each σ was measured using growth assays across a range of inductions in LB media (Figure 3C; Supplementary Figure S8), and with colony size on LB agar plates (Supplementary Table S2.4). Essentially none of the σs were toxic (<75% *E. coli* DH10b carrying an empty vector) during exponential phase in liquid media even at 100 μM induction (99% non-toxic), and only a fraction of the σs exhibited toxicity at transition phase in liquid media or on plates (77 and 90% non-toxic at 100 μM induction, respectively). The two ECF σs from subgroup 02 exhibited the highest toxicity. Subgroup 02 includes *E. coli* $\sigma^{E}$ (candidate ECF02_2817), which is known to be toxic when highly expressed (Nitta *et al*, 2000); consequently, the toxic effects of high expression of both ECF02 σ members in the library (ECF02_2817 and ECF02_915) suggest a similar function. Two less toxic σs (ECF20_992 and ECF34_1384) were simultaneously expressed at high induction to test whether toxicity would increase with co-expression. Transition phase measurements of growth in LB media showed that the cells with both σs were no worse than those expressing only ECF20_992 (Supplementary Figure S11).

**Figure 3** The activity and orthogonality of ECF σs are shown. (**A**) ECF σs are induced by IPTG via a T7 expression system, and σ-dependent promoter activity was measured by *gfp* expression and flow cytometry. Plasmid pN565 (incW ori) encodes the IPTG-inducible T7* expression system (Temme *et al*, 2012); plasmid series pVRa (pBR322 ori) and pVRb (pSC101 ori) encode the ECF σ library and test promoter library, respectively. The specific example shown (ECF11_987 and P$_{11\_3726}$) is highlighted in the following subfigures. (**B**) Activities of active ECF σ library members titrated against their target promoters. The gray lines show levels of GFP expression for one active ECF σ:promoter pair in each subgroup induced with 0, 10, 20, 50, and 100 μM IPTG. The averaged activity of σ ECF11_987 against its promoter P$_{11\_3726}$ is highlighted in black. Data are shown from three independent assays and error bars represent one standard deviation. Plots of the other σ:promoter pairs are shown in more detail in Supplementary Figure S4. (**C**) The liquid culture growth curves (OD$_{600}$) are shown for each σ under high induction (100 μM IPTG). The growth curve of σ ECF11_987 averaged from three independent growth assays is highlighted in black and the error bars represent one standard deviation. Background growth curves show data from one growth assay. The growth curves of two negative controls are shown in dark gray. Note that 64 out of the 86 σs show no growth impact as compared with the control. (**D**) The activity of one promoter (P$_{11\_3726}$) is shown for the complete library of active σs expressed with 100 μM IPTG. Each bar represents the average promoter activity from at least two independent assays and error bars represent one standard deviation. The two σs from subgroup 11 that were expected to activate the promoter are bracketed. (**E**) All cross reactions are shown for the 20 most orthogonal σ:promoter pairs. Each σ is induced with 100 μM IPTG, and the fold induction is measured as the fluorescence with σ induction divided by the basal activity of the promoter in the absence of any σ. Each square represents the average fold induction from at least two independent assays of a unique σ:promoter combination. All promoters were named using the convention P$_{\mathbf{XX\_YYYY}}$, where 'XX' and 'YYYY' denote the subgroup and unique ID of the downstream parent σ gene (e.g., P$_{02\_2817}$ is the promoter upstream of σ ECF02_2817). Promoters containing synthetic UP elements were renamed to P$_{\mathbf{XX\_UPYYYY}}$ (e.g., P$_{15\_UP436}$). The σ:promoter pairs were ordered by the absolute amount of off-target activity caused by/ affecting the pair, with the lowest off-target activity in the upper left and the highest in the lower right. (**F**) Promoter scores, as calculated from PWMs, are compared with the experimental measurements in (**E**). The promoter scores are calculated using the ECF promoter models (UP + PWM$_{−35}$ + PWM$_{−10}$ + spacer penalty) for the − 60 to + 20 promoter fragment including 30 nt flanking vector sequence. The ECF11_987:P$_{11\_3726}$ activity is highlighted in red. (**G**) ECF02_2817 and ECF11_3276 were recombined in their flexible linker region between domains 2 and 4 to create chimeric σs ECF02-11 and ECF11-02. The promoters activated by the two parental σs were similarly recombined between the − 10 and − 35 regions to create chimeric promoters. (**H**) The activity and orthogonality of the σs and chimeric σs are shown against their cognate promoters. All of the σs are induced with 10 μM IPTG and the fold induction is as defined previously. Each square represents the average fold induction from three independent assays.

All promoters were then assayed against the complete set of σs. This resulted in an exhaustive activity map based on the measurements of 29 promoters against 86 σs (2494 data points), which was performed in duplicate for the 27 most functional promoters. An example of a screen for one promoter against all 86 σs is shown in Figure 3D, and the full duplicate data set is provided in Supplementary Table S2.1. Twenty-six active promoters were identified from this screen (Supplementary Figure S4). In total, 58 of the 86 σs activated at least one promoter >5-fold. Among the inactive σs, usually both examples from a subgroup were non-functional, suggesting that their promoter motifs were incorrect or a shared property of these σs prevents function in *E. coli*. The transfer

function was measured for each of the 52 most active σs against its most active promoter (Figure 3B; Supplementary Figure S5).

As predicted by the promoter models, many of the σ/ promoter pairs are highly orthogonal and the 20 most orthogonal are shown in Figure 3E. Some crosstalk is consistently observed between different subgroups (ECF02, 07, 11, 14, 15, 17, 25, 27, 33). These off-target interactions are due to subgroups sharing similar promoters; thus, they can be predicted using the promoter models (Figure 3F).

In addition, to determine the orthogonality of σ/promoter pairs, it is important to determine whether the σ factors alter transcription of the host cell genome. To investigate this, we

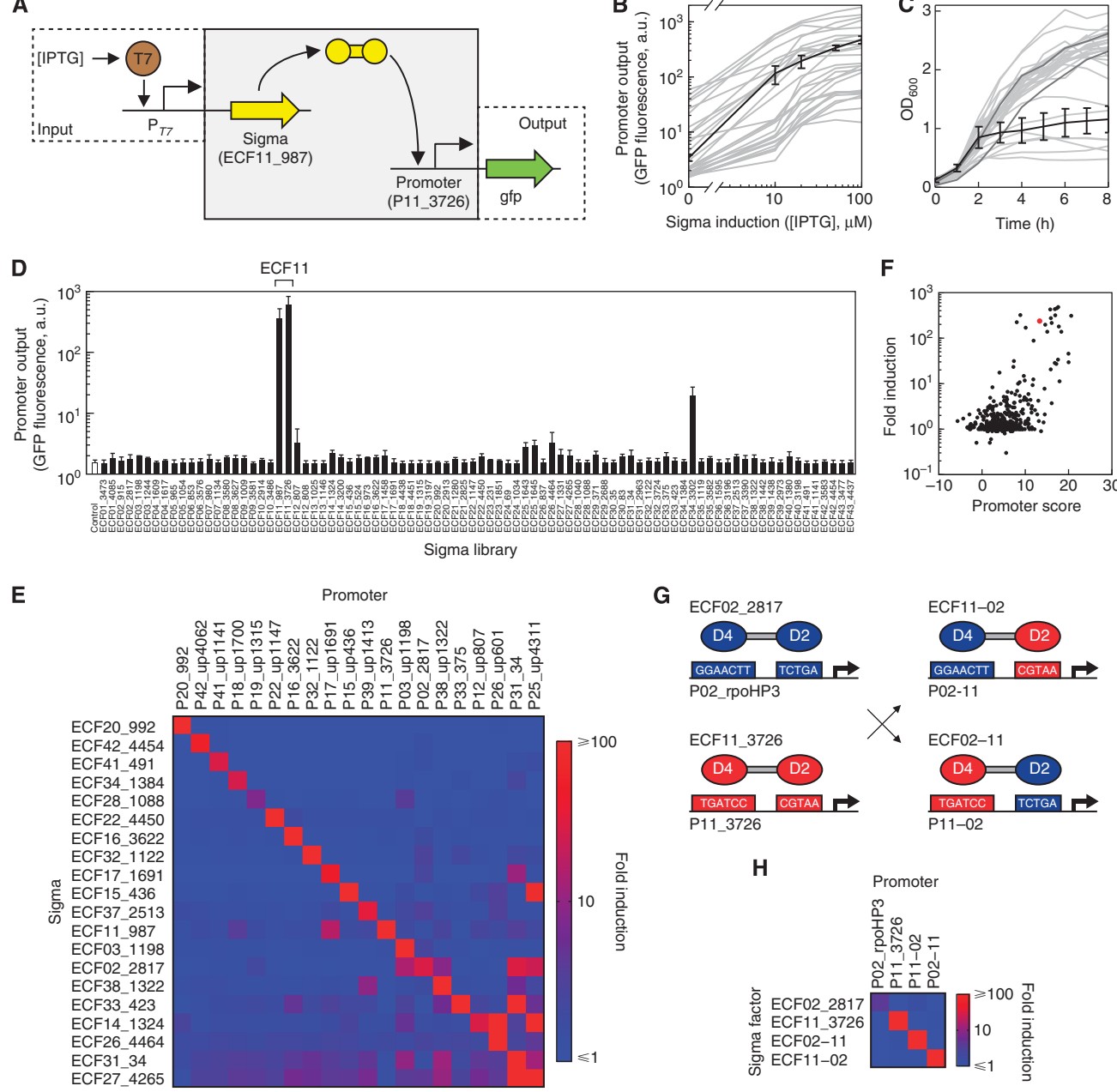

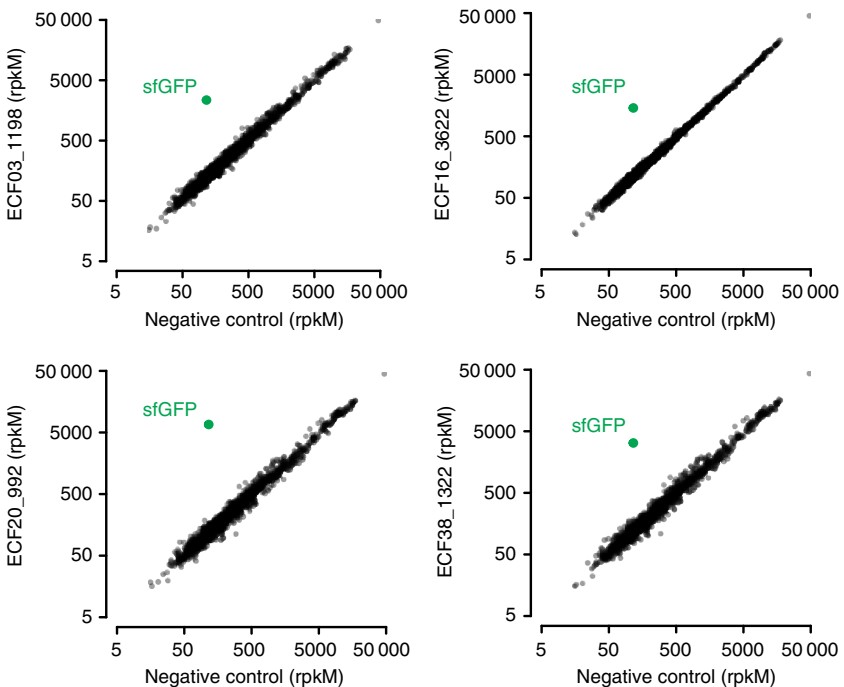

**Figure 4** Overexpression of ECF σs has minimal off-target effects on the host genome. ECF σs from groups 03, 16, 20, and 38 are induced with 20 μM IPTG via the T7 expression system detailed above, and genome-wide transcription is measured by RNA-seq. Transcription of *sfgfp* from cognate ECF promoters encoded on the appropriate pVRb-series plasmids is also measured. As a negative control lacking exogenous σs, a strain carrying pET21a in place of a pVRa plasmid, as well as pVRb03_up1198, is used. Transcription of *sfgfp* is induced 10–60 fold in all samples relative to the negative control strain, while changes in host genome transcription are minimal. Sequencing read counts are quantified as reads per kilobase per million reads (rpkM), which is adjusted for CDS length and total sequencing reads.

over-expressed σs from groups ECF 03, 16, 20, and 38 and used RNA-seq to compare their genome-wide expression profiles with that of a control strain lacking these σs (Figure 4). These σs were chosen because they were highly orthogonal to other σs and also had partner anti-σs that were reasonably orthogonal (see below), interacting primarily with their partner σ. These properties make these σs particularly useful for construction of large synthetic circuits. Although all four exogenous σ factors induced transcription of GFP from their cognate promoters 10–60-fold, no significant changes in expression from the host genome were detected. This finding underscores the high information content of ECF σ promoters, and suggests that complex synthetic circuits utilizing multiple exogenous ECF σs can be constructed without significant off-target transcriptional effects on the host genome.

## Chimeric σs with swapped −10 and −35 motif recognition

The promoter models demonstrate that the promoter specificities of ECF σs are a product of the combination of the −10 and −35 motifs, which are recognized by two separate domains of σ (Supplementary Figure S2), and further indicate that not all of the potential combinations of −35 and −10 sequences are represented in the 43 subgroups (Figure 2). Thus, if a chimeric σ could be constructed by combining the −10 binding domain from one σ with the −35 binding domain from another σ, this could enable the combinatorial design of new σs that could dramatically increase the number

of available orthogonal σs. We roughly predict that across subgroups, 16 σ domains bind to different −35 sequences, and 10 σ domains bind to different −10 sequences. Thus, considering only DNA sequence specificity, we estimate that there is an upper limit of ∼160 potential ECF σs that could be orthogonal and potentially operate within one cell.

Indeed, a synthetic hybrid σ combining the −10 DNA binding domain of $\sigma^{70}$ with the −35 DNA binding domain of $\sigma^{32}$ was able to recognize a cognate hybrid promoter containing a consensus $\sigma^{32}$ −35 motif and a consensus $\sigma^{70}$ −10 motif (Kumar *et al*, 1995), suggesting the feasibility of constructing chimeric ECF σs to increase the diversity of orthogonal ECF σs and their promoters.

The ECF σs are simple, consisting of two domains separated by a flexible linker. The N-terminal domain (domain 2) binds the −10 motif and the C-terminal domain (domain 4) binds the −35 motif (Campbell *et al*, 2003, 2007; Lane and Darst, 2006). We tested whether these domains could be swapped between different ECF subgroups to create chimeric σs that activate chimeric promoters. Two σs were selected from different subgroups that recognize different −10 and −35 motifs: ECF02_2817 (*E. coli* $\sigma^E$) and ECF11_3726 (Figure 3G). For each orientation, six chimeric σs were constructed by making crossovers in the disordered linker region and in helixes near the domain boundary (Supplementary Figure S18). Similarly, a library of three chimeric promoters was constructed based on the −35 and −10 motifs from $P_{02\_rpoHP3}$ and $P_{11\_3726}$ that represent a range of spacer lengths. From these small libraries, chimeric σs were identified that activate their chimeric promoters at a level equivalent to wild type (Figure 3H;

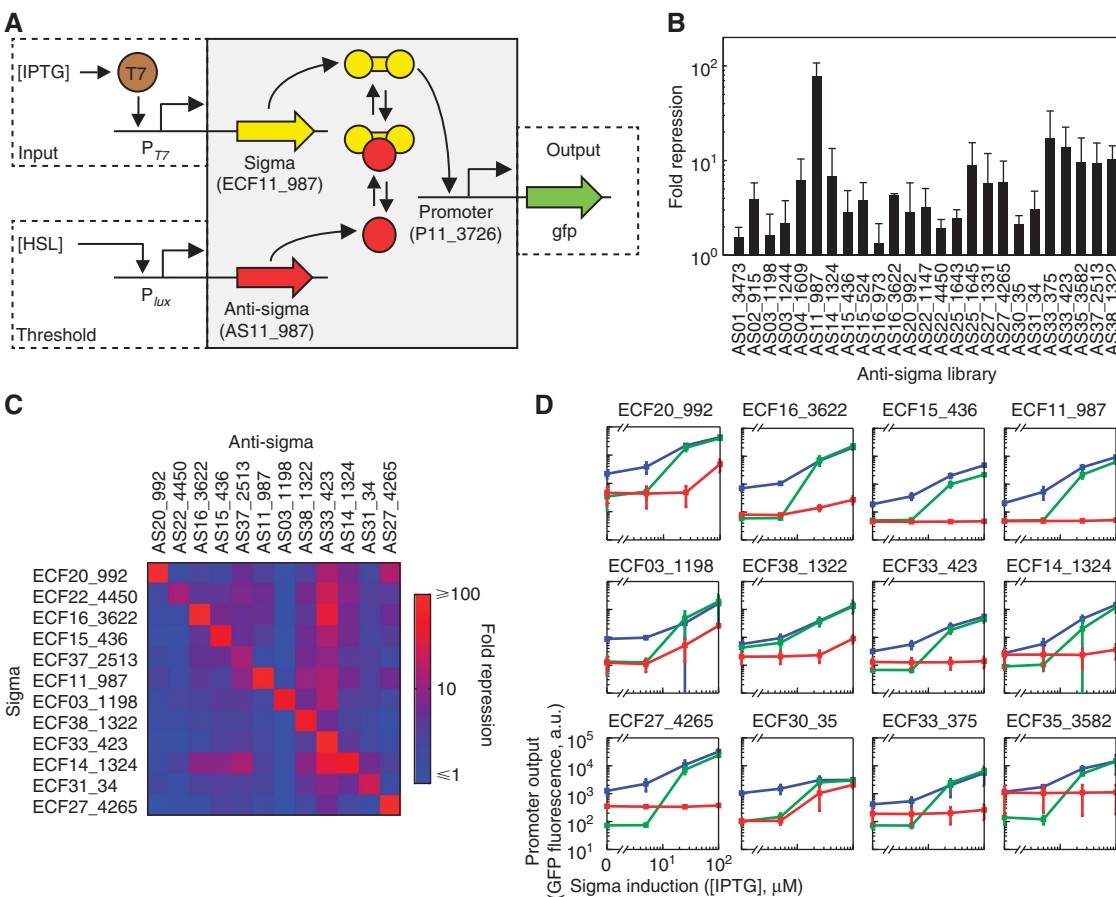

**Figure 5** Anti-σs can be used to create orthogonal threshold-gated switches. (**A**) In addition to the expression and reporter systems shown in Figure 3A, cells contain the plasmid series pVRc (p15A ori), which allows HSL-inducible independent expression of anti-σs to bind and sequester σs. (**B**) Repression of ECF11_987 activity on promoter $P_{11\_3726}$ by different anti-σs. Each bar represents average fold repression, as defined by normalizing the fluorescence of cells containing the promoter with both induced σ (induced with 10 μM IPTG) and induced anti-σ (induced with 50 nM HSL) against the fluorescence of cells containing just the promoter and induced σ. Bar heights represent the average from at least two independent assays and error bars represent one standard deviation. (**C**) The crossreactivity of 12 anti-σs on the set of 12 orthogonal σs targeted by the anti-σs. The activity of each σ paired with its cognate promoter was measured in the absence and presence of different anti-σs. σs were partially induced (10 μM IPTG) and anti-σs maximally induced (50 nM HSL). Colors indicate fold activity repression by the anti-σ, defined as the activity of the promoter in the absence of the anti-σ divided by the activity in its presence. The anti-σ:σ pairs were arranged in the same order as the σs in Figure 3E. Each square represents the average of at least two independent assays. (**D**) The influence of the expression of the anti-σ is shown for a series of switches. The plots show expression of GFP from each σ-dependent promoter for differing expression levels of cognate anti-σ induced by the inducer HSL: blue, no anti-σ plasmid; green, 10 nM HSL; red, 50 nM HSL. The data represent the average of three independent assays and error bars show standard deviations. Representative cytometry distributions are shown in Supplementary Figure S15.

Supplementary Figures S19–S21). The ECF02 and ECF11 σs activate their promoters 4.1-fold and 169-fold, respectively, and their chimeras activate the chimeric promoters 107-fold and 151-fold. The chimeric σs are also orthogonal, exhibiting negligible activity against the opposite promoter chimeras and the parental promoters. Interestingly, while *E. coli* σ$^E$ (ECF02_2817) is toxic at high concentrations (Nitta *et al*, 2000), neither of the chimeras based on this σ exhibit toxicity (Supplementary Figure S22).

## Mining anti-σ factors and the orthogonality of their protein–protein interactions with σs

Anti-σs bind to σs and inhibit them by blocking their interaction with core RNAP. Using Staroń's list of ECF σs and their cognate anti-σs, we constructed a library of 62 anti-σs that were cognate to σs in our library (Figure 1B; Supplementary Table S1.3). Of these, 46 anti-σs were

associated with an active σ. Using the 35 most active of these σs paired with their most active promoter, we tested whether their cognate anti-σs were able to repress activity by expressing them from a 3-O-C$_6$-HSL inducible P$_{lux}$ promoter (Figure 5). Out of this set, 32 anti-σs were able to repress the activity of their target σ by >2-fold (Supplementary Figure S6), indicating that most of the anti-σs from different organisms were able to repress their target σ in *E. coli*.

Compared with the σs, a larger fraction of the anti-σs exhibited toxicity when expressed in *E. coli* (Supplementary Figure S9; Supplementary Table S2.5). This could occur by the interaction of the anti-σs with essential host σs such as ECF σ$^E$. At high levels of expression (50 nM HSL), 83% of the anti-σs exhibited near wild-type growth (>75%) during exponential phase in LB media, and only 49 and 46% did not fall below that threshold at transition phase in LB media or as judged by colony size on LB agar plates, respectively.

To determine the orthogonality of the anti-σ/σ interactions, the activity of 36 of the σs was measured in the presence of the 25 most active anti-σs. This screen yielded a matrix with $25 \times 36 = 900$ data points (Supplementary Table S2.3). As an example, a row is shown in Figure 4B where the anti-σ AS11_987 represses the activity of its cognate σ ECF11_987 by over 70-fold whereas the other anti-σs have a significantly smaller effect. Measurements of repression by all anti-σs against all of their cognate σs reveal that most anti-σs specifically repress their cognate σ, but there is a higher background of crossreactivity against non-cognate σs (Supplementary Figure S7). Figure 5C shows a subset of this data for the 20 most orthogonal σs (Figure 3E) for which functional anti-σs were identified.

### Threshold-gated synthetic genetic switches

To demonstrate how the orthogonal σs, anti-σs, and promoters could be used to build synthetic circuits, we assembled them into transcriptional switches. This is a relatively simple motif that could form the basis of more complex dynamical functions, such as bistable switches (Chen and Arkin, 2012), pulse generators (Basu *et al*, 2004), or oscillators (Stricker *et al*, 2008). Transcriptional circuits are defined as having inputs and outputs that are both promoters, thus facilitating their connection to build more complex functions. The σs and their promoters could be used to build transcriptional switches (Figure 3A), where an input promoter (e.g., $P_{T7}$) drives the σ and the σ-dependent promoter serves as the output. However, the majority of σs yield a graded induction of the output promoter (Figure 3B). This induction occurs at a similar threshold of input activity. There is also a 100-fold range in the basal OFF state in the presence of uninduced σ. All of these features pose a challenge for assembling the switches into larger programs: more complex dynamical behaviors require a cooperative response, connecting circuits requires matching their thresholds, and a high basal level can trigger the next circuit in series.

Previously, it was demonstrated that the addition of a sequestering molecule into a switch lowers the background, increases the cooperativity and allows the threshold to be tuned (Buchler and Cross, 2009). We tested whether the anti-σs serving as a sequestering molecule would improve these properties of the switches. A series of switches was constructed based on controlling the expression level of anti-σ (Figure 5A; Supplementary Figure S23). In each case, as the expression level of the anti-σ increased, the basal level of expression decreased (up to 10-fold) and the cooperativity increased (Figure 5D). For example, when detailed data for ECF20_992 are fit to a Hill function, the cooperativity goes from $n = 1.7$ to 4.1 as a function of the expression of the anti-σ (Supplementary Figure S17). There was also an exquisite capacity for the threshold of the circuit to be tuned, in several cases, by over two orders of magnitude.

## Discussion

Using genome mining, we have created a core orthogonal set of 20 σ:promoter pairs that exhibit minimal crosstalk, and a set of complementary anti-σs that downregulate only their cognate σ. This represents a major addition to the synthetic biology parts list. These parts functioned off-the-shelf in an *E. coli* host with minimal re-engineering, and minimally affected host growth and gene expression. This underscores the value of the ever-expanding pool of sequenced genomes as a toolbox for synthetic biology.

A significant outcome of this work is the ease with which ECF σs can be moved between diverse organisms and retain function. The σs used for construction of our library were obtained from across six different bacterial classes, but their functionality was unrelated to phylogenetic distance from *E. coli*. For example, non-functional σs were observed from γ-proteobacteria, the same subclass as *E. coli*, and functional σs were observed from Firmicutes, the most distant class. Transferability of σs across widely divergent bacteria probably results from the fact that RNAP subunits are reasonably well conserved across bacteria (Lane and Darst, 2010; Werner and Grohmann, 2011), and that the physical interface between σ and RNAP is extensive (Sharp *et al*, 1999; Murakami *et al*, 2002). This work implies that these ECF σs will be able to function in many host backgrounds, including the wide range of prokaryotic species increasingly being used for engineering novel production circuits, and potentially even chloroplasts. To probe the transferability of ECF σs, one σ from our library was moved into *Klebsiella oxytoca*, where it functioned comparably to *E. coli* (Supplementary Figure S12).

The toxicity that was observed for a small fraction of the heterologous ECF σs could derive from competition for native RNAP with host σs and/or from aberrant gene expression. *E. coli* $\sigma^E$ and other members of subgroup 02 were the most toxic ECF σs. At least in this case, toxicity is likely to arise from overexpression of particular regulon members (Nitta *et al*, 2000; Asakura and Kobayashi, 2009). $\sigma^E$ monitors and responds to outer membrane stress by increasing the expression of proteins that facilitate production and assembly of outer membrane components, and by inducing sRNAs to shut down synthesis of outer membrane porins and other proteins that need assembly in the envelope compartment of the cell. Aberrant overexpression of the components leads to imbalanced growth and sometimes to lysis, the likely source of toxicity (Nitta *et al*, 2000). More generally, we note that ECF σs are likely to have lower affinity for RNAP than other σ groups because they lack domain 3, which contains some RNAP binding determinants (Murakami and Darst, 2003). Indeed, $\sigma^E$ has one of the lowest affinities for RNAP of any σ (Maeda *et al*, 2000). The ability of these heterologous σs to direct gene expression despite poor affinity for RNAP is probably a consequence of their high target promoter specificity and low non-specific DNA binding activity compared with σ 70 (Grigorova *et al*, 2006).

Another important outcome of this work is the demonstration that orthogonality is a combinatorial property of the − 10 and − 35 promoter regions, encoded respectively by domains 2 and 4 of the ECF σ. We first demonstrated this computationally (Supplementary Figure S2), showing that either promoter region alone was significantly less orthogonal than the complete promoter, and followed this by an experimental demonstration that we could mix and match ECF σ domains to achieve new, orthogonal recognition. In addition to its obvious

biotechnological implications, this finding is important in considering ECF σ evolution. The fact that many of the ECF σs are orthogonal suggests that diversifying the combinatorial output was a key consideration in evolution of these proteins.

Finally, this work provides the first overview of how bacterial genomes organize ECF σ transcriptional space. As ECF σs are the most abundant σs, with genomes encoding as many as 50–60 of these transcriptional regulators, partitioning of this transcriptional space is a key consideration in the types of ECF σs encoded in a genome (Helmann, 2002). σs that do not crosstalk with each other enable insulated expression of various regulons, whereas crosstalking ECF σs potentially allow multiple σs to cooperate in coordinated responses (Mascher *et al*, 2007). When insulation of existing pathways is crucial, selective pressures will favor acquisition of non-crosstalking ECF σs (Capra *et al*, 2010). As a first step in this analysis, we have analyzed the co-occurrence of non-crosstalking versus crosstalking σs in bacterial genomes. The 11 most non-crosstalking ECF σs have on average almost twice as many co-occurring σ partners compared with the 12 crosstalking σs (22 versus 12; $P = 0.034$; Supplementary Table S3.1). This analysis suggests that it is usually beneficial for cells to insulate their gene expression pathways, leading to the preferential co-occurrence of non-crosstalking σs. As one example, of the 8 ECF σs in *C. crescentus*, 6 are orthogonal, 1 crosstalks, and the last has not been tested. Additionally, some of the most orthogonal ECF σs (e.g., groups 41 and 42) are widely distributed, whereas the most promiscuous (ECF25) shows much more limited distribution. It will be of interest to determine whether these highly orthogonal σs retain particular regulon functions across bacteria, or are rewired to perform new functions.

On the other hand, cells can take advantage of crosstalking σs to build complex regulatory networks. A good example is provided by the work of John Helmann on the ECF σs in *B. subtilis* (Helmann, 2002; Mascher *et al*, 2007; Luo *et al*, 2010). All of the ECF σs in this organism crosstalk and all respond to envelope stress of various sorts (Luo *et al*, 2010). As a result, *B. subtilis* has been able to construct a complex response network in which some core genes induced by all stresses, and an additional set of genes is induced by only one or several stresses. It will be important to determine whether the co-occurrence of crosstalking ECF σs generally signals cooperation in responding to important signals, or whether organisms have taken further steps at the genome level to insulate these σs. Finally, our analysis reveals another possible mechanism of σ–σ collaboration, possibly leading to hierarchical expression. In several instances an ECF σ does not autoregulate, but co-occurs in a genome with an ECF σ that can transcribe its upstream promoter, raising the possibility of a σ–σ cascade (Supplementary Table S3.4). Such cascades could orchestrate complex decisions, akin to the *B. subtilis* sporulation σ cascade (Stragier and Losick, 1990).

Further analysis of our data is likely to prove useful in addressing these and other biological questions, and for reverse engineering of ECF signaling pathways in general. Additionally, these data are likely to stimulate direct experimentation to test our insights into genome scale organization of these transcriptional circuits. Finally, our data set highlights the valuable role of forward-engineering approaches in facilitating understanding of natural biological circuits.

# Materials and methods

## Strains and media

*Escherichia coli* strain DH10β (MC1061 F-endA1 recA1 galE15 galK16 nupG rpsL ΔlacX74 Φ80lacZΔM15 araD139 Δ(ara,leu)7697 mcrA Δ(mrr-hsdRMS-mcrBC) λ-) (Durfee *et al*, 2008) was used for all manipulations and assays unless otherwise noted. *E. coli* DH10β strains were grown at 37°C with aeration in LB Miller broth for expression assays, and in LB Miller broth, 2YT, SOB (2% Bacto-tryptone, 0.5% Bacto yeast extract, 10 mM NaCl, 2.5 mM KCl), SOB + Mg (SOB + 10 mM MgCl₂, 10 mM MgSO₄) or SOC (SOB + Mg + 20 mM glucose) for cloning and CaCl₂ high-throughput transformations. *E. coli* strain CAG22216 (MC1061λ(rpoH P3-lacZ) rpoE::Cam, CamR) (Rouvière *et al*, 1995) was used for expression and testing of chimeric σs. *E. coli* CAG22216 strains were grown at 30°C with aeration in the same media as *E. coli* DH10β. *Klebsiella oxytoca* strain M5a1 (Bao *et al*, 2013) was used to test σ transferability. *K. oxytoca* was grown at 30°C with aeration in the same media as *E. coli* DH10β. All cultures were supplemented with appropriate antibiotics. Expression of the σ library and chimeric σs was induced with 0–100 μM IPTG (from Sigma Aldrich, #I6758). The anti-σ library was induced with 0–50 nM HSL (3-O-C₆-HSL (*N*-(β-ketocaproyl)-L-Homoserine Lactone from Cayman Chemical, #10011207)). Cultures were grown in 96-well format in sterile V96 tissue culture plates (NUNC, cat #249935) using either an ELMI plate shaker-thermostat (DTS-4 from Elmi Ltd, Riga, Latvia) shaking at 1000 r.p.m. at 37 or 30°C or a Multitron Pro (Infors HT, Bottmingen, Switzerland) shaking at 900 r.p.m. at 37°C. Plates were covered with gas-permeable membranes (AeraSeal from EK Scientific, cat #T896100-S).

## High-throughput transformations of σ and anti-σ libraries

*In vivo* assays of strains carrying σ or anti-σ libraries were performed from freshly transformed *E. coli* DH10β host cells. This was to reduce the occurrence of potential suppressor mutations from toxicity of some of the σs and anti-σs by long-term maintenance in a host. A CaCl₂-based high-throughput transformation protocol in 96-well format was employed that enabled convenient transformation of several hundred strains a day. CaCl₂ competent cells were prepared using the method of Hanahan *et al* (1991) for MC1061-based strains. Briefly, 50 ml cultures of cells were grown in SOB ( − Mg) media, harvested at OD₆₀₀ = 0.3, pelleted and supernatant discarded, cells resuspended and pelleted in 25 ml ice-cold CaCl₂ buffer (50 mM CaCl₂, 10 mM Tris–HCl pH 7.5) and then finally resuspended in 3.3 ml fresh ice-cold CaCl₂ buffer + 15% glycerol. Plasmid DNA stocks of each library were prepared at 5 ng/μl in 96-well format. For transformation, 10 ng of each plasmid was placed into a sterile 96-well PCR plate with 25 μl ice-cold CaCl₂ competent cells and incubated on ice for 60 min (for double plasmid transformations, 5 ng each plasmid + 40 μl CaCl₂ competent cells was used). The entire PCR plate was then heat-shocked at 42°C in a dry block for 2 min and then placed on ice for 5 min. Afterwards, cells were transferred to a fresh 96-well tissue culture plate containing 100 μl SOC, mixed, sealed with a breathable membrane and incubated at 37°C, 1000 r.p.m. for 2 h. In all, 30 μl cells were then transferred to a fresh 96-well tissue culture plate containing 130 μl SOB + Mg + appropriate antibiotics for selection, covered with breathable membrane and incubated overnight (~16 h) at 37°C, 1000 r.p.m. This liquid selection in the presence of antibiotics was sufficient to prevent growth of no plasmid controls. The fresh overnight transformants grown to saturation were used for all downstream assays by diluting 200-fold into fresh media with antibiotics and inducers, and growing fresh cultures as required.

## σ activity assays

The σ-promoter gfp assays were performed in *E. coli* DH10β host cells using a 3 plasmid system: pN565 carrying IPTG-inducible T7 RNAP, pVRa plasmid series carrying the σ library, and pVRb plasmid series carrying σ promoters fused to sfgfp (Supplementary Figure S23).

Titrations of σ against a specific promoter (Figure 3B; Supplementary Figure S5) were performed at 0, 5, 10, 20, 50, and 100 μM IPTG. Assays of all σs against all promoters (Figure 3E; Supplementary Figure S4) were also performed in 96-well format with each plate containing the entire σ library assayed against a specific promoter. Specifically, *E. coli* DH10β cells carrying pN565 and a specific pVRb promoter::sfgfp plasmid were transformed with the complete pVRa σ library and pET21a control in 96-well format. Overnight liquid transformants grown to saturation (∼16 h) were diluted 200-fold into fresh prewarmed LB + Spec, Amp, Kan and 100 μM IPTG in a 96-well cell-culture plate and covered with a breathable membrane. Cultures were incubated in an Elmi plate shaker for 6 h at 37°C, 1000 r.p.m. After 6 h, 5 μl of culture was added to 200 μl PBS (137 mM NaCl, 2.7 mM KCl, 8 mM Na$_2$HPO$_4$, 2 mM KH$_2$PO$_4$) and 2 mg/ml Kanamycin. Samples were run on a BD Biosciences LSRII flow cytometer to quantify GFP accumulation.

### Anti-σ library activity assays

The anti-σ activity assays were performed in *E. coli* DH10β host cells using a 4 plasmid system: pN565 carrying IPTG-inducible T7 RNAP, pVRa plasmid series carrying the σ library, pVRb plasmid series carrying σ promoters fused to sfgfp, and pVRc plasmid series carrying the anti-σ library under HSL-inducible control. Plasmid pACYC184 was used as a no anti-σ control. Anti-σ activity was determined by its ability to repress σ activity. Accordingly, each σ was paired with its most active promoter to determine fold repression in the presence and absence of anti-σ. The anti-σ activity assays were performed exactly as described for the σ-promoter assays with the following differences:

#### Anti-σ–σ titration assays

In Supplementary Figure S6, the transfer functions are shown for the σ for different levels of anti-σ expression. For each anti-σ–σ titration set, *E. coli* DH10β cells carrying pN565 and a specific pVRc anti-σ were doubly transformed with pVRa σ/pVRb promoter plasmid pair. A single overnight transformation was then diluted 200-fold into 12 wells of a 96-well plate containing LB, Spec, Amp, Kan and Cm, and a 2-dimensional grid of inducer concentrations: 0, 5, 20, or 100 μM IPTG, and 0, 10, or 50 nM HSL. A no anti-σ control was also included using DH10β pN565 pACYC184 cells doubly transformed with pVRa σ/pVRb promoter plasmid pair. The control was diluted 200-fold into 4 wells of a 96-well plate containing LB + Spec, Amp, Kan, and Cm and 0, 10, 20, or 100 μM IPTG.

#### Anti-σ–σ activity assays

Twenty-five anti-σs were assayed against 36 active σs paired with an active promoter in 96-well format (Figure 5C; Supplementary Figure S7). To maximize the ability of anti-σs to repress target σs, the σs were only partially induced with 10 μM IPTG and the anti-σs maximally induced with 50 nM HSL. DH10β cells carrying pN565 and a specific pVRc anti-σ (or no anti-σ control, pACYC184) were doubly transformed with a library of pVRa σ/pVRb promoter plasmid pairs. Overnight liquid transformants were diluted 200-fold into fresh prewarmed LB + Spec, Amp, Kan, 10 μM IPTG and 50 nM HSL.

### σ and anti-σ exponential phase liquid growth rate assays

These were performed by diluting freshly transformed overnight cultures 200-fold into prewarmed LB media with appropriate antibiotics and inducer. Cultures were in 96-well cell-culture plates covered with a clear lid and were grown in a Varioskan plate reader/shaker (Thermo Fisher Scientific) at 37°C, shaking at 480 r.p.m., 6 mm orbital motion. Cell densities (OD$_{600}$) were recorded automatically by the Varioskan every 15 min for 2 h during exponential growth. All OD$_{600}$ measurements on the Varioskan were converted to standard 1 cm pathlength ODs using a calibration curve generated from an

exponentially growing 50 ml culture in a 250-ml shake flask. Samples from the shake flask were taken every hour throughout the growth curve and the OD$_{600}$ measured in a 1 cm pathlength cell with a standard spectrophotometer (with appropriate dilution so that OD$_{600}$ readings were always between 0.25 and 0.35) and from 160 μl samples measured in a 96-well cell-culture plate by the Varioskan. The calibration curve generated from the plot of actual 1 cm pathlength OD$_{600}$ values versus 160 μl Varioskan OD$_{600}$ values was used to normalize all experimental culture ODs measured in the Varioskan. The normalized experimental OD readings were plotted as $\ln(OD_{600})$ versus time (h). Bacterial growth rate μ was calculated from the slope of the linear section of the plot,

$$\mu = (\ln N_t - \ln N_0)/(t - t_0),$$

where μ is the growth rate, $N$ is the number of cells (approximated by OD), and $t$ is the time. The growth rates of all σ and anti-σ libraries were expressed as a percentage of WT (averaged from eight control cultures).

### Transition phase liquid cell densities

These were performed exactly as the exponential phase growth rate assays with the following modifications. Assay cultures were induced and grown in the Varioskan for 8 h and the growth curve monitored from OD$_{600}$ readings performed every hour. Wild-type cultures typically entered transition phase after 2–3 h. Sick cultures often exhibited a decrease in culture OD$_{600}$ values during transition phase, likely due to cessation of growth and subsequent cell lysis. Transition phase cell densities were recorded from the final 8 h OD$_{600}$ values, normalized to 1 cm pathlength ODs and presented as a percentage of WT OD$_{600}$ (from eight control cultures).

### Colony size measurements

These were performed in 96-colony format from 96-well cultures. Fresh overnight transformants in 96-well format from each library were pinned onto separate LB-agar master plates containing appropriate antibiotics using a Singer Rotor robot and a 96-pin liquid to solid pinner head. Each plate was incubated overnight for 14 h at 37°C to grow colonies in 96-format. From each master plate, colonies were pinned onto inducer plates with the Singer robot using 96-pin solid to solid pinners. The inducer plates contained LB agar plus appropriate antibiotics and IPTG or HSL inducer, and were incubated overnight for 14 h at 37°C to grow colonies in 96-format. Colony sizes were recorded using a 6 megapixel camera under controlled lighting (Typas *et al*, 2008), and colony diameter measured using the automated image analysis software, HT Colony Grid Analyzer (http://sourceforge.net/projects/ht-col-measurer/files/). The sizes of all σ and anti-σ expressing colonies were converted to a percentage of WT (from two control colonies).

### Anti-σ threshold assays

The anti-σ:σ titrations were repeated in more detail (Figure 4D; Supplementary Figure S10). DH10β cells were transformed with pN565 and a set of pVRa, pVRb, and pVRc plasmids corresponding to one of 16 promising anti-σ: σ: promoter sets. A negative anti-σ control for each set was also made that lacked the pVRc plasmid. Glycerol stocks were made of each strain and stored at −80°C. For each assay, the glycerol stocks were used to start overnights in LB + Amp, Spec, Kan, and Chl (or Amp, Spec, and Kan for the no anti-σ controls). After growing to saturation, these overnights were diluted 1:200 into LB + antibiotics in a 96-well cell-culture plate. The four plasmid strains were added to a grid of inducer concentrations: 0, 5, 20, or 100 μM IPTG, and 0 or 50 nM HSL. The no anti-σ strains were added to inducer conditions of 0 nM HSL and 0, 5, 20, or 100 μM IPTG. Each 96-well plate was shaken for 6 h at 37°C, 1000 r.p.m. In all, 2 μl of each induction was added to 198 μl PBS + 2 mg/ml Kanamycin and stored at 4°C. Samples were run on a BD Biosciences LSRFortessa flow cytometer to quantify GFP accumulation.

## Chimeric σ assays

Assays of chimeric σ function were performed in both delta *rpoE* (ECF02_2817) *E. coli* strain CAG22216 (Rouvière *et al*, 1995) (Supplementary Figures S14 and S15) and DH10β (Figure 3H; Supplementary Figure S16) supplemented with antibiotics as required. A two plasmid assay system was used, consisting of the series pTSaXX, which contains the parental and chimeric σs under IPTG-inducible control, and pTSbXX, which contains the parental and chimeric σ promoters driving *sfgfp* (Supplementary Figure S17).

Similar to the assays with σ or anti-σ libraries, the chimeric σs were transformed into cells directly before assaying. Z-competent (Zymo Research, cat# T3002) cell stocks of *E. coli* CAG22216 or *E. coli* DH10β carrying plasmids from series pTSbXX were made as per manufacturer's instructions. The day before the functional assay, 100 ng aliquots of plasmids from series pTSa were added to 50 μl Z-competent cells at 4°C. The cells were kept on ice for 10 min, 100 μl SOC was added and the cells were outgrown at 30°C, 1000 r.p.m. (*E. coli* CAG22216) or 37°C, 900 r.p.m. (*E. coli* DH10β) for 2 h. These growths were diluted 1:100 into 150 μl LB + Spec/Kan, and incubated overnight (∼16 h) under the same conditions as for the outgrowth.

For both *E. coli* CAG22216 assays, transformed overnights were diluted 1:200 into LB + Spec/Kan + 10 μM IPTG and grown for 8 h at 30°C, 1000 r.p.m. In all, 5 μl of each induction was added to 195 μl PBS + 2 mg/ml Kanamycin and stored at 4°C. Samples were run on a BD Biosciences LSRFortessa flow cytometer to quantify GFP accumulation. For the DH10β orthogonality assay, the protocol was the same, except transformed overnights were diluted 1:100 into LB + Spec/Kan + 10 μM IPTG and grown for 6 h at 37°C, 900 r.p.m.

For the chimeric σ growth assays, transformed overnights were diluted 1:200 into LB + Spec/Kan + 10 μM IPTG, grown for 8 h at 37°C, 1000 r.p.m., and the $OD_{600}$ was measured in a Synergy H1 plate reader (BioTek Instruments Inc., Winooski, VT). Growth assays were performed in cells with a negative control reporter plasmid, pTSb01. All data were normalized to the OD of cells containing an empty σ expression plasmid as a negative control (pTSa01), and is presented as a percentage of that value.

## Flow cytometry analysis

GFP fluorescence of the diluted samples was measured using either a BD Biosciences LSRII flow cytometer (UCSF) or a BD Biosciences LSRFortessa flow cytometer (MIT). Initial analysis of the σ and anti-σ libraries was performed on the LSRII, while the rest of the analysis, including threshold and chimera testing, was done with the LSRFortessa.

### LSRII analysis

For each sample, 50 000 counts were recorded using a 0.5 μl/s flow rate. All data were exported in FCS2 format and processed using FlowJo (TreeStar Inc., Ashland, OR). Data were gated by forward and side scatter and the geometric mean fluorescence was calculated.

### LSRFortessa analysis

For each sample, at least 5000 counts were recorded using a 0.5 μl/s flow rate. All data were exported in FCS3 format and processed using FlowJo (TreeStar Inc.). Data were gated by forward and side scatter then gated to remove any fluorescence values lower than 0. The geometric mean fluorescence was calculated from this gated population.

### Fold calculations

Promoter activity represents the mean fluorescence value obtained from flow cytometry analysis. Fold induction is calculated by measuring the mean fluorescence for cells containing a σ and a reporter and dividing it by the activity of cells containing the reporter but not the σ. Similarly, fold repression is calculated by measuring the mean fluorescence of cells containing the σ and the reporter and dividing by the mean fluorescence of cells containing the anti-σ, the σ, and the reporter.

## mRNA sequencing

Cultures for mRNA sequencing were prepared using a modification of the σ-promoter gfp reporter assay protocol. Overnight liquid transformants were diluted 200-fold into 25 ml fresh prewarmed LB + Spec, Amp, Kan, and 20 μM IPTG in 125-ml flasks. Cultures were grown in a water bath shaker to $OD_{600} \sim 0.35$ at 37°C, 240 r.p.m. Total RNA was phenol chloroform extracted from lysates, and small RNAs and rRNA were subtracted with MEGAclear and MICROBExpress kits (Ambion). The mRNA samples were fragmented using RNA fragmentation reagents (Ambion), and fragments were converted to cDNA libraries as described previously (Ingolia *et al*, 2012). Sequencing was performed on an Illumina HiSeq 2000 system. Sequencing reads were mapped to the *E. coli* DH10β reference genome NC_010473.1 using Bowtie v.1.0.0. For all samples, at least two million sequencing reads mapped to CDS regions. Data were analyzed with Python and R scripts. Analyses included the 1421 most highly expressed genes, below which noise dominated the signal due to low molecular counts.

## Supplementary information

## Acknowledgements

CAV is supported by Life Technologies, DARPA CLIO (N66001-12-C-4018), Office of Naval Research N00014-10-1-0245, NIH AI067699, and the NSF Synthetic Biology Engineering Research Center (SynBERC, SA5284-11210). CAG is supported by NIH R01GM057755-31. THSS is supported by a National Defense Science and Engineering Graduate Fellowship and by a Fannie and John Hertz Foundation Fellowship. BDS and DHB are supported by NIH T32GM008284.

*Author contributions:* CAV, THSS, VR, CAG, AG, BDS, DHB, EO, and HT designed and performed the experiments, analyzed the data, and wrote the manuscript. TP and KC aided project management, performed the DNA synthesis, and edited the manuscript.

## Conflict of interest

This research has been sponsored by Life Technologies and there is a relevant patent application.

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
