## [Review Process File · Molecular Systems Biology]

Design of orthogonal genetic switches based on a crosstalk map of σ s, anti- σ s, and promoters

Virgil R. Rhodius, Thomas H. Segall-Shapiro, Brian D. Sharon, Amar Ghodasara, Ekaterina Orlova, Hannah Tabakh, David H. Burkhardt, Kevin Clancy, Todd C. Peterson, Carol A. Gross, Christopher A. Voigt

Corresponding author: Christopher A. Voigt, Massachusetts Institute of Technology

Review timeline:

Submission date:	06 May 2013
Editorial Decision:	19 June 2013
Revision received:	17 September 2013
Accepted:	26 September 2013

Editor: Maria Polychronidou

Transaction Report:

1st Editorial Decision

19 June 2013

Thank you again for submitting your work to Molecular Systems Biology. We have now heard back from the three referees who agreed to evaluate your manuscript. As you will see from the reports below, the reviewers acknowledge the potential value of your work for synthetic biology applications. However, they raise a series of concerns, which should be carefully addressed in a revision of the manuscript.

Without repeating all the points listed below, one of the more fundamental issues pointed out by reviewer #1, refers to the need to perform further experimentation to determine the specificity of the foreign sigma factors, when induced in *E. coli*. Along the same lines, reviewer #3 mentions that the potential cross-talk of heterologous sigma factors/ anti-sigma factors/ promoters with native, host components should be addressed. Additionally, the referees require additional explanations and/or clarifications for several points throughout the manuscript.

On a more editorial note, as Reviewer #3 has pointed out, the potential significance of the work for synthetic biology applications should be clearly stated in the abstract.

If you feel you can satisfactorily deal with these points and those listed by the referees, you may wish to submit a revised version of your manuscript. Please attach a covering letter giving details of the way in which you have handled each of the points raised by the referees. A revised manuscript will be once again subject to review and you probably understand that we can give you no guarantee at this stage that the eventual outcome will be favorable.

REFeree REPORTS:

Reviewer #1:

This paper presents experimental data regarding the cross-specificity of different ECP sigma factors selected from the main bacterial groups. This information could be extremely valuable for building new synthetic circuits in bacteria. The results are very clearly presented. However I have two major concerns that should be addressed before the paper is accepted.

- 1) I could not find the expression level of the sigma factors at different IPTG concentrations compared to cognate sigma ECPs, as well as the plasmid copy number of the reporter. If these circuits should be used in synthetic biology, then 1-5 copy plasmid should be used and expression levels of the sigma ECPs should be within physiological limits.
- 2) At a time where deep sequencing for bacteria is cheap and fast, the authors should check the possible changes in expression of *E. coli* genes upon induction by IPTG of the foreign sigmas at concentrations where they activate the reporter. This could help in determining how specific they are and how many off targets they have.

Reviewer #2:

In this manuscript, Rhodiu et al. have identified and characterized a library of ECF sigma factors and ECF sigma factor related components to regulate gene expression in *Escherichia Coli*. The authors expand on previous efforts on identifying ECF sigma factors by developing a novel computational strategy for identifying a cognate promoter sequence associated with an ECF sigma factor. Using their method, the authors develop ECF sigma factor open reading frames whereby a particular ECF sigma factor can up regulate gene expression from its cognate promoter and where, if it exists, an anti-ECF can repress gene expression from the same promoter *in vivo*. The authors also characterize the functional orthogonality and the toxicity of each component in *E. coli*, noting that some components are functionally orthogonal to each other while not being toxic to *E. coli*. Finally, the authors engineer novel ECF sigma factors and ECF sigma factor promoters by mixing and matching -35 and -10 DNA binding domains to create unique sets of ECF sigma factor open reading frames. In summary, this work presents library components that have been in high demand in the field while suggesting strategies for generating identifying and engineering additional ECF sigma factor components.

This paper represents a significant contribution to the field by presenting an exciting new resource for researchers in the field. While the idea of importing components from distant hosts into *E. coli* is not new, ECF sigma factors and anti-sigma factors *in vivo* opens up new and unique opportunities for tunable gene expression *in vivo*. The authors should also be applauded for meticulously reporting their experimental conditions and methodologies such that readers can better interpret the data. However, while the broad claims of this project are supported by the data without doubt, parts of the study need to be made consistent within itself for a more cogent document. I recommend publication provided the following points are addressed:

1. GFP fluorescence (Au) reported as a proxy for promoter activity. As there is not yet an agreed upon metric for promoter activity, the authors should report the output of their reporter constructs as GFP fluorescence rather than promoter activity. While both units are arbitrary measures and distinctions, at least GFP fluorescence is what is directly measured. Promoter activity cannot be directly substituted in for GFP fluorescence in this case as the authors have not taken measures to insulate their reporter constructs from local effects such as variations in the 5'UTR.
2. It is not clear why the authors assert that promoter activity measurements were taken during the exponential growth phase in 6hrs when in fact according to the growth curves graphed in figure 3C, a majority of the growth curves have in fact reached late exponential or stationary phase. The authors should simply note promoter activity was measured after 6 hrs without reference to the growth phase.
3. Furthermore, the growth curves plotted in figure 3C should be on a linear scale to be more in line with what is reported in literature and to better characterize the effect of expressing ECF sigma factors may have in a cell.
4. Non-preliminary data that is referenced in the main body of the text should be at least from three experimental replicates with error bars representing the standard deviation. This is lacking in figure

3D and 3E to ensure reproducibility and for an understanding of the expected variance when these components.

5. Figure S10 should be referenced in the caption of figure 4D such that the bimodal nature of some of the populations with anti-sigma repression can be better observed.

6. If data on synthetic sigma factors is to be included with data on the orthogonal set of ECF sigma factors, then the experiment should at least be done in a similar fashion. ECF02_2817, ECF02_3726, ECF02-11, ECF11-02 should be tested against the subset of promoter libraries in DH10B cells with 100uM IPTG. At the very least, the authors should adopt a different coloring scheme to emphasize the fact that the data in 3H is from an entirely different experiment than in parts a-e. Furthermore, the chimeric sigma factors cannot be claimed to be functionally orthogonal in the same way as the assayed ECF sigma factors unless the experimental systems are demonstrated to be equivalent to one another.

7. It is not clear what HSL concentrations were used on page 12 of the supplement when the authors are detailing (0,50, and 100nM HSL) should that range of inducer concentrations be (0,10,50 nM HSL) instead?

8. Data in figure S6 should be done in triplicate, especially if it is discussed in the main text.

9. Figure S3, each bar should have the same number of experimental replicates or it should be noted the number of experimental replicates associated with each bar.

10. To gain a better understanding of the level of induction of the test systems, transfer functions of the promoters used to drive expression of the components in their experimental setups should be provided. This will help readers gauge how close to saturation the expression systems used in this study are operating.

Reviewer #3:

This manuscript has two main threads: (i) the authors combine bioinformatics and gene expression assays to survey the extent of specificity and crosstalk between ECF sigma factors, anti-sigma factors, and promoters across bacteria and (ii) the authors use this screen to identify orthogonal sigma/anti-sigma pairs for synthetic biology purposes in *E. coli*.

We felt that the authors could better situate their work. Their bioinformatics is similar to Staron et al, *Mol. Microbiol.* (2009) who first analyzed and classified ECF sigma and anti-sigma factors (see comment #1 below). The main difference is that Voigt and colleagues experimentally measure specificity and crosstalk of sigma, anti-sigma, and promoters. Bio prospecting for orthogonal sigma/anti-sigma pairs is an exciting direction. The synthetic biology field has been limited by a handful of natural TFs (LacI, TetR, AraC, LuxR, etc..). Zinc fingers and TAL effectors are one route for designing novel TFs. However, these designer TFs often lack cooperativity, a necessary ingredient for generating thresholds, bistability, and oscillations. Recent work shows that titration of TFs (e.g. sigma factor) by inhibitors (e.g. anti-sigma factors) can generate sharp thresholds. Voigt and colleagues screened 43 ECF subtypes down to ~12 useful sigma / anti-sigma pairs. The authors then demonstrated that these sigma / anti-sigma pairs generate threshold responses. Thus, they have uncovered an orthogonal sigma / anti-sigma toolkit that might accelerate the development of synthetic genetic switches and oscillators. This is significant and novel, but you would never know it from their title or abstract.

There remains one challenge that dampens our optimism for this synthetic biology toolkit. Unwanted interactions between these heterologous sigma/anti-sigma factors with host sigma / anti-sigma factors can lead to mis-regulated gene expression (and toxicity). This 'squelching' is especially relevant because sigma factors directly bind and compete for RNAP holoenzyme and mis-regulate global gene expression. The authors were careful to measure toxicity via different growth assays, and indeed, some sigma/anti-sigma pairs displayed acute toxicity in *E. coli*. However, the authors only studied one pair at a time -- it is likely that several sigma/anti-sigma pairs together will be acutely toxic. This lack of scaling will undermine the usefulness of this molecular toolkit. We think it is important for the authors to demonstrate that multiple sigma/anti-sigma pairs can be

tolerated by *E. coli* and not exhibit unwanted interactions through RNAP or native sigma/anti-sigmas (see comments #5-7 below).

Comments:

1. The authors have a refined PWM approach that finds twice as many putative sigma sites (29 versus 16) than Staron et al. The authors should compare and contrast their bioinformatic results (e.g. position weight matrices) to those of Staron et al. Did both approaches agree? Why or why not?
2. How were two subgroup members for each ECF subfamily chosen? Was it arbitrary? Or were two members chosen for maximum or minimum phylogenetic divergence? We could not easily find this information in SOM or the main text.
3. For a given ECF sigma factor, it was unclear whether the authors paired it with the putative promoter for the same sigma factor in the exact same genome. Please elaborate.
4. We were surprised that sigma overexpression is not more toxic. Why is it not toxic? Are heterologous ECF sigmas generally weaker binders to native holoenzyme when compared to native housekeeping sigmas? Please discuss and cite supporting references.
5. The authors compare heterologous sigma/anti-sigma/promoter to each other. However, we think they should also test specificity and cross-talk with native, host sigma / anti-sigma / promoters from *E. coli*. For example, Figure S4 should include *E. coli* ECF and *E. coli* promoters associated with ECF (and other sigmas). This would clearly demonstrate the extent to which heterologous ECFs affect host gene expression (and host-specific cross-talk that can lead to toxicity).
6. Another important control would be to transform plasmids into a different host bacterium. Are results of Figures 3-4 independent of host organism, as expected?
7. Unless the authors also design an anti-sigma(s) to their chimeric sigma(s), there is no gained advantage. As such, chimeric sigma data in Fig. 3g-h is distracting.
8. When compared to Staron et al, why do you not have a PWM for ECF 04, 13, 23, 28, 34-36, 43 in Fig. S1?
9. The authors state that Figure S2 shows that "-10 and 35 sequences alone show considerably less orthogonality than the entire promoter". By eye, the difference between -10, -35, and both does not seem that significant ... can you quantify?
10. Because the matrix is not symmetric, it is hard to "see the diagonal" in Fig. S4. It would be useful to have white boxes bracketing ECF and Promoter pixels that are expected to interact (i.e. the diagonals) in Fig. S4.
11. Bimodal GFP distribution in Fig. S5 may be correlated with toxicity, where the low GFP fraction consists of faster-growing sigma suppressors that are accumulating in the cell culture. If so, how should we interpret those bimodal GFP distributions with no obvious toxicity phenotype (hatched red lines, as measured by your growth assays)?
12. Figure S6 has colors with no explanation. What is red, black, grey? What are the controls?
13. Toxicity data is missing. Where is "colony size on LB agar plate" data?

Minor comments:

14. Unclear how and why sigma-70 PWM was used in this research? It is described in SOM, but its connection to your research is missing.
15. On pg. 7 in SOM, "Many non-functional promoter constructs had poor upstream sequences with AAA and TTT-tract counts of <2". Awkward sentence seems to imply that A/T tracts are poor upstream sequences. Rewrite.

We have made the changes suggested by the reviewers, including significant new experimentation. Some major changes include:

- We have added RNA-seq experiments (Figure 4) to show the orthogonality of the sigma factors with respect to crosstalk with the *E. coli* genome. Note that there is remarkably little crosstalk.
- We have added experiments showing the functionality of the sigmas in other bacteria (*Klebsiella*) (Supplemental Section II.G.).
- We have added experiments showing the impact on toxicity of expressing multiple sigmas (Supplemental Section II.F.).
- As requested by the Editor and Reviewer 3, we have changed the title and abstract to focus on the impact in synthetic biology.
- New experiments showing the induction curves for T7 RNAP driven sigma expression were performed (Figure S14).
- We repeated the synthetic chimera experiments so that the data is in the same strain background as the sigma measurements. This did not have a significant impact on the data.
- The manuscript text has been significantly edited and put into the format and constraints of an MSB article.

The detailed changes that occurred as a result to each reviewer comment are listed below.

Reviewer #1:

1. *I could not find the expression level of the sigma factors at different IPTG concentrations compared to cognate sigma ECFs, as well as the plasmid copy number of the reporter. If these circuits should be used in synthetic biology, then 1-5 copy plasmid should be used and expression levels of the sigma ECFs should be within physiological limits.*

Indeed, we agree that lower copy number plasmids are preferred. We are working with a multi-plasmid system. The expression system is based on a low-copy (~3 copies/cell) plasmid and a medium copy (15-20 copies/cell) plasmid. We selected this system because it tightly controls T7 RNAP expression. The information regarding the origins was in the SI, but we agree that it should be in the main text as well and have included it now in the figure captions. We also have performed new experiments showing the expression level of the sigma factors at different IPTG concentrations (Figure S14).

2. *At a time where deep sequencing for bacteria is cheap and fast, the authors should check the possible changes in expression of E. coli genes upon induction by IPTG of the foreign sigmas at concentrations where they activate the reporter. This could help in determining how specific they are and how many off targets they have.*

Rna-seq experiments have been added to the manuscript (Figure 4). We picked several sigma factors, including one that exhibited some toxicity in *E. coli*. Remarkably, almost no crosstalk with the host genome is observed.

Reviewer #2:

1. *GFP fluorescence (Au) reported as a proxy for promoter activity. As there is not yet an agreed upon metric for promoter activity, the authors should report the output of their reporter constructs as GFP fluorescence rather than promoter activity. While both units are arbitrary measures and distinctions, at least GFP fluorescence is what is directly measured. Promoter activity cannot be directly substituted in for GFP fluorescence in this case as the authors have not taken*

measures to insulate their reporter constructs from local effects such as variations in the 5'UTR.

We have changed the axis labels able to read: Promoter Output (GFP fluorescence, au).

2. *It is not clear why the authors assert that promoter activity measurements were taken during the exponential growth phase in 6hrs when in fact according to the growth curves graphed in figure 3C, a majority of the growth curves have in fact reached late exponential or stationary phase. The authors should simply note promoter activity was measured after 6 hrs without reference to the growth phase.*

We have removed the references to growth phase as suggested.

3. *Furthermore, the growth curves plotted in figure 3C should be on a linear scale to be more in line with what is reported in literature and to better characterize the effect of expressing ECF sigma factors may have in a cell.*

We have changed the figure to be on a linear plot. We note that the majority of the curves are within or above the cell controls containing no sigma factor (64 out of 86 sigmas or 75%).

4. *Non-preliminary data that is referenced in the main body of the text should be at least from three experimental replicates with error bars representing the standard deviation. This is lacking in figure 3D and 3E to ensure reproducibility and for an understanding of the expected variance when these components.*

Obtaining the data in this paper was a remarkable feat – each replicate for the full data set of sigma:promoter pairs is 2,376 data points (not including anti-sigma measurements). We view this as a screen that we performed in duplicate (4,752 data points), which was a real tour-de-force. The data in Figures 3D and 3E are showing one example of a particular sigma from this much larger data set. Keep in mind that the purpose of this screen is just to identify putative crosstalk. Once we identified the core set of orthogonal sigmas, this data was measured in triplicate or more over multiple levels of induction (Figures 3B and S5).

5. *Figure S10 should be referenced in the caption of figure 4D such that the bimodal nature of some of the populations with anti-sigma repression can be better observed.*

We have added this reference to the caption of Figure 4D (Now 5D).

6. *If data on synthetic sigma factors is to be included with data on the orthogonal set of ECF sigma factors, then the experiment should at least be done in a similar fashion. ECF02_2817, ECF02_3726, ECF02-11, ECF11-02 should be tested against the subset of promoter libraries in DH10B cells with 100uM IPTG. At the very least, the authors should adopt a different coloring scheme to emphasize the fact that the data in 3H is from an entirely different experiment than in parts a-e. Furthermore, the chimeric sigma factors cannot be claimed to be functionally orthogonal in the same way as the assayed ECF sigma factors unless the experimental systems are demonstrated to be equivalent to one another.*

We have repeated this measurement in *E. coli* DH10B cells to ensure that the different strain was not leading to incomparable results with the rest of the data in the paper. This new assay is shown in Figure 3H and Figure S16. Changing strains does not have an effect on the function of the chimeric sigma factors or the level of crosstalk between the chimeras and parents.

7. *It is not clear what HSL concentrations were used on page 12 of the supplement when the authors are detailing (0,50, and 100nM HSL) should that range of inducer concentrations be (0, 10, 50 nM HSL) instead?*

Yes, and this has been corrected.

8. *Data in figure S6 should be done in triplicate, especially if it is discussed in the main text.*

This is similar to the comment in point 4. We performed the large-scale screen, requiring 560

datapoints, from which we pulled the most interesting parts that are then reported in the main text with more replicates. While the data generated from this screen is not done in replicate, we feel it is valuable to show it in the SI. We note that, while it is mentioned, no claims in the main text are dependent on this data set.

9. *Figure S3, each bar should have the same number of experimental replicates or it should be noted the number of experimental replicates associated with each bar.*

We have added this information to the caption.

10. *To gain a better understanding of the level of induction of the test systems, transfer functions of the promoters used to drive expression of the components in their experimental setups should be provided. This will help readers gauge how close to saturation the expression systems used in this study are operating.*

We have performed new experiments to measure the transfer functions, which are reported in Supplemental Section II.H. In all cases, we are operating well under the maximal levels of induction possible with these systems.

Reviewer #3:

Thus, they have uncovered an orthogonal sigma / anti-sigma toolkit that might accelerate the development of synthetic genetic switches and oscillators. This is significant and novel, but you would never know it from their title or abstract.

We have edited the title, abstract, and main text (especially the discussion) to increase the emphasis on the impact on synthetic biology.

*There remains one challenge that dampens our optimism for this synthetic biology toolkit. Unwanted interactions between these heterologous sigma/anti-sigma factors with host sigma / anti-sigma factors can lead to mis-regulated gene expression (and toxicity). This 'squelching' is especially relevant because sigma factors directly bind and compete for RNAP holoenzyme and mis-regulate global gene expression. The authors were careful to measure toxicity via different growth assays, and indeed, some sigma/anti-sigma pairs displayed acute toxicity in *E. coli*. However, the authors only studied one pair at a time -- it is likely that several sigma/anti-sigma pairs together will be acutely toxic. This lack of scaling will undermine the usefulness of this molecular toolkit. We think it is important for the authors to demonstrate that multiple sigma/anti-sigma pairs can be tolerated by *E. coli* and not exhibit unwanted interactions through RNAP or native sigma/anti-sigmas (see comments #5-7below).*

We have performed several new experiments to further analyze the possibility of crosstalk and squelching. We have added RNA-seq experiments to measure the impact of expressing the sigmas on gene expression across the *E. coli* genome (Figure 4). Essentially no crosstalk is observed. As suggested, we also co-express sigma factors to determine whether extra toxicity due to squelching is observed (Supplemental Section II.F.). We do not see particularly strong toxicity to occur when multiple sigmas are expressed.

Comments:

1. *The authors have a refined PWM approach that finds twice as many putative sigma sites (29 versus 16) than Staron et al. The authors should compare and contrast their bioinformatic results (e.g. position weight matrices) to those of Staron et al. Did both approaches agree? Why or why not?*

The promoter motifs identified by both ourselves and Staron are very similar. We have included more information detailing this comparison in the Supplemental Information. The Staron analysis was based on the identification of at least 10 upstream regions of ECF operons (start to -250) and known promoters on which the single block motif algorithm MEME was used to search for motifs.

In brief, we used several new strategies to maximize the success of identifying promoter sequences:

- We used BioProspector (Liu et al. 2001), which is a 2-block motif search algorithm that is ideally suited for bacterial promoters with variable length spacers between the -10 and -35 motifs. Consequently, this algorithm is more sensitive than MEME.
- Most promoters occur near the start of genes, but some can be located far upstream. To maximize the success of identifying promoter motifs, different length upstream regulatory sequences (from the start codon to 100, 150, 200 and 300 nt upstream) were extracted for each library.
- We utilized the regulatory sequences of all σ genes within each subgroup. This increases the ability to find poorly conserved motifs or motifs that are only present in a subset of sequences. We then searched for over-represented promoter motifs directly upstream of the: i) σ gene; ii) 1st gene of the operon containing the σ gene. In the case of the operons there are many instances where the σ gene is internal to the operon and the auto-regulatory promoter was identified upstream of the entire operon.

2. *How were two subgroup members for each ECF subfamily chosen? Was it arbitrary? Or were two members chosen for maximum or minimum phylogenetic divergence? We could not easily find this information in SOM or the main text.*

We have added a new section in the SI to describe the choice of the ECF subfamily members (Section II.A. Selecting ECF sigma factors, anti-sigma factors, and promoters). In brief, to maximize phylogenetic diversity, 2 σ s were selected from each of the 43 ECF subgroups defined by Staron *et al.* to create a library of 86 σ s. Within each subgroup, σ s were preferentially selected from genomes closely related to *E. coli* to maximize the likelihood of binding to *E. coli* RNAP. Since some ECF subgroups only contain σ s from genomes phylogenetically distant to *E. coli* this still resulted in a σ library spanning 6 bacterial classes. σ s were also selected if they had a known cognate anti- σ (Staron et al. 2009).

3. *For a given ECF sigma factor, it was unclear whether the authors paired it with the putative promoter for the same sigma factor in the exact same genome. Please elaborate.*

We have added material to Supplemental Section II.A to clarify this point. When possible, a given sigma was paired with the putative promoter from the same genome. In these cases, the promoter and ECF sigma factor have the same unique ID (e.g., ECF02_2817 and P02-2817). However, this pairing was not always possible and several criteria were used in selecting the final promoters for each sigma group:

- Preference was given to promoters that were predicted to be orthogonal against the other ECF σ s: *i.e.* scored highly in their own promoter model and scored poorly against the other promoter models.
- Promoters were also screened against any overlapping host promoter sequences using an *E. coli*-specific σ^{70} promoter model for the housekeeping σ and the ECF05-10 promoter model for Fecl. This was especially important for promoters selected from A/T-rich genomes, since they often contained weak overlapping σ^{70} promoter signals that are also A/T-rich.

4. *We were surprised that sigma overexpression is not more toxic. Why is it not toxic? Are heterologous ECF sigmas generally weaker binders to native holoenzyme when compared to native housekeeping sigmas? Please discuss and cite supporting references.*

We have added a paragraph on this topic to the discussion, including references. To our knowledge, there is no extensive thermodynamic study of ECF sigma binding to their native holoenzyme and it is not possible to draw conclusions from the little data that is available. This makes it difficult to interpret the toxicities based on competing binding affinities. Note also that the majority of sigmas in our library are not native to *E. coli* and it may be that their affinity for *E. coli* holoenzyme is much lower than their native polymerase. Whilst they are still able to bind *E. coli* RNAP, evolutionary divergence most likely weakens this binding, enabling the native sigmas to still compete. The ability of the heterologous sigmas to still direct gene expression is probably a consequence of their high target promoter specificity and low non-specific DNA binding activity compared to sigma 70.

5. *The authors compare heterologous sigma/anti-sigma/promoter to each other. However, we think they should also test specificity and cross-talk with native, host sigma / anti-sigma / promoters from E. coli. For example, Figure S4 should include E. coli ECF and E. coli promoters associated with ECF (and other sigmas). This would clearly demonstrate the extent to which heterologous ECFs affect host gene expression (and host-specific cross-talk that can lead to toxicity).*

We have added new RNA-seq experiments (Figure 4) to the text to measure the crosstalk between the sigmas and all *E. coli* promoters.

6. *Another important control would be to transform plasmids into a different host bacterium. Are results of Figures 3-4 independent of host organism, as expected?*

We have transformed a sigma factor into another host organism (*Klebsiella*) and obtained similar results (Supplemental Section II.G.).

7. *Unless the authors also design an anti-sigma(s) to their chimeric sigma(s), there is no gained advantage. As such, chimeric sigma data in Fig. 3g-h is distracting.*

We have edited the text to clarify the importance of Figure 3g-h. One of the conclusions of this work is that the -10 and -35 binding domains of the σ independently contribute to the binding and neither is sufficient on its own. This was implied by the promoter modeling experiments and experimentally verified by the chimeras that we built. Because the -10 and -35 domains can be crossed, this dramatically increases the number of potential sigma factors that could be constructed beyond the 43 subgroups that have been observed in nature.

8. *When compared to Staron et al, why do you not have a PWM for ECF 04, 13, 23, 28, 34-36, 43 in Fig. S1?*

These ECF groups were relatively small and only contained 10-30 sigmas (except ECF43, which contains 36 sigmas). This makes it inherently difficult to identify motifs by searching for conserved over-represented sequence, especially if the promoters are poorly conserved. In the upstream regions of several ECF groups, several strong “non ECF-like” motifs were observed, some of which were palindromic, suggesting that they were transcription factor binding sites. The presence of these strong motifs biases the search and reduces the ability of algorithms like Bioprospector from finding other, less significant motifs. Some sigma groups are highly phylogenetically related; consequently there is insufficient sequence diversity within the upstream regulatory regions to find over-represented motifs. In these cases, the highly similar sequences were removed from the search; however, there were often too few remaining sequences from which to identify conserved motifs. It is also possible that the promoters of some of these sigmas are very poorly conserved, are located outside the search windows, or that these sigmas simply do not autoregulate.

9. *The authors state that Figure S2 shows that "-10 and 35 sequences alone show considerably less orthogonality than the entire promoter". By eye, the difference between -10, -35, and both does not seem that significant ... can you quantify?*

We have modified Figure S2 to show the Z-score analysis of the promoters and subsites as a way to compare the relative predicted off-target effects. The Z-score is a normalized promoter score, where the raw promoter model score is normalized using the predicted on-target promoter scores to show the number of on-target standard deviations any promoter is from the on-target mean score (See the end of section I.B. and equation S4). With this analysis, the increase in off target effects when only using the -10 or -35 models is readily apparent. Additionally, paired t-tests performed between the off-target z-scores showed that there is a significant difference between the full model and both the -10 and -35 models ($p < 0.05$). This analysis has been added to the Supplemental Information.

10. *Because the matrix is not symmetric, it is hard to "see the diagonal" in Fig. S4. It would be useful to have white boxes bracketing ECF and Promoter pixels that are expected to interact (i.e. the diagonals) in Fig. S4.*

White boxes have been added around the predicted cognate ECF sigma / promoter pairs in this figure.

11. *Bimodal GFP distribution in Fig. S5 may be correlated with toxicity, where the low GFP fraction consists of faster-growing sigma suppressors that are accumulating in the cell culture. If so, how should we interpret those bimodal GFP distributions with no obvious toxicity phenotype (hatched red lines, as measured by your growth assays)?*

We have included additional supplementary data with all of the growth data. The majority of biomodal plots do exhibit at least some growth defect. For the few that remain, mild toxicity may not always perturb the growth rate under our conditions, but could more easily affect cell morphology. This could be reflected in the scatter plots, which are a function of cell shape and granularity.

12. *Figure S6 has colors with no explanation. What is red, black, grey? What are the controls?*

We have edited the caption to include this information.

13. *Toxicity data is missing. Where is "colony size on LB agar plate" data?*

We have added supplementary tables that include this information. Supplementary Table 2 has assay data for the σ and anti- σ libraries, including the missing toxicity data. Specifically, toxicity data for the σ s is in Supplementary Table S2.4 and toxicity data for the anti- σ s is in S2.5.

14. *Unclear how and why sigma-70 PWM was used in this research? It is described in SOM, but its connection to your research is missing.*

The sigma-70 PWM was used to scan promoters to screen putative ECF sigma promoters to see if they might have an overlapping promoter sequence that is active in *E. coli*. We have added more discussion of this including Section II.A. in the Supplemental Information.

15. *On pg. 7 in SOM, "Many non-functional promoter constructs had poor upstream sequences with AAA and TTT-tract counts of <2". Awkward sentence seems to imply that A/T tracts are poor upstream sequences. Rewrite.*

We have edited this sentence for clarity.

Acceptance letter

26 September 2013

Thank you again for sending us your revised manuscript. We are now satisfied with the modifications made and I am pleased to inform you that your paper has been accepted for publication.

Thank you very much for submitting your work to Molecular Systems Biology.

Reviewer #3 (Report):

The authors have done a good job comparing their results to previous work and addressing our concerns regarding toxicity and off-target effects.